# Oregano (*Origanum vulgare*) Essential Oil and Its Constituents Prevent Rat Kidney Tissue Injury and Inflammation Induced by a High Dose of L-Arginine

**DOI:** 10.3390/ijms25020941

**Published:** 2024-01-11

**Authors:** Nikola M. Stojanović, Katarina V. Mitić, Milica Nešić, Milica Stanković, Vladimir Petrović, Marko Baralić, Pavle J. Randjelović, Dušan Sokolović, Niko Radulović

**Affiliations:** 1Department of Physiology, Faculty of Medicine, University of Niš, 18000 Niš, Serbia; pavleus@gmail.com; 2Institute of Physiology and Biochemistry “Ivan Djaja”, Faculty of Biology, University of Belgrade, 11000 Belgrade, Serbia; katarinavmitic@yahoo.com; 3Department of Chemistry, Faculty of Sciences and Mathematics, University of Niš, 18000 Niš, Serbia; milica.stevanovic992@gmail.com (M.N.); nikoradulovic@yahoo.com (N.R.); 4Department of Pathology, Faculty of Medicine, University of Niš, 18000 Niš, Serbia; stankovic.milica93@gmail.com; 5Department of Histology and Embryology, Faculty of Medicine, University of Niš, 18000 Niš, Serbia; vlada@medfak.ni.ac.rs; 6School of Medicine, University of Belgrade, 11080 Belgrade, Serbia; baralicmarko@yahoo.com; 7Department of Nephrology, University Clinical Centre of Serbia, 11000 Belgrade, Serbia; 8Institute for Biochemistry, Faculty of Medicine, University of Niš, 18000 Niš, Serbia; dusantsokolovic@gmail.com

**Keywords:** kidney, L-arginine, *Origanum vulgare*, inflammatory parameters, tissue damage parameters

## Abstract

This study aimed to evaluate the protective action of oregano (*Origanum vulgare*) essential oil and its monoterpene constituents (thymol and carvacrol) in L-arginine-induced kidney damage by studying inflammatory and tissue damage parameters. The determination of biochemical markers that reflect kidney function, i.e., serum levels of urea and creatinine, tissue levels of neutrophil-gelatinase-associated lipocalin (NGAL), and kidney injury molecule-1 (KIM-1), as well as a panel of oxidative-stress-related and inflammatory biomarkers, was performed. Furthermore, histopathological and immunohistochemical analyses of kidneys obtained from different experimental groups were conducted. Pre-treatment with the investigated compounds prevented an L-arginine-induced increase in serum and tissue kidney damage markers and, additionally, decreased the levels of inflammation-related parameters (TNF-α and nitric oxide concentrations and myeloperoxidase activity). Micromorphological kidney tissue changes correlate with the alterations observed in the biochemical parameters, as well as the expression of CD95 in tubule cells and CD68 in inflammatory infiltrate cells. The present results revealed that oregano essential oil, thymol, and carvacrol exert nephroprotective activity, which could be, to a great extent, associated with their anti-inflammatory, antiradical scavenging, and antiapoptotic action and, above all, due to their ability to lessen the disturbances arising from acute pancreatic damage. Further in-depth studies are needed in order to provide more detailed explanations of the observed activities.

## 1. Introduction

In rats, the injection of a single high dose of L-arginine, a basic amino acid, causes pancreatic tissue damage associated with inflammatory cell infiltration, tissue edema, and necrosis [1,2]. The progression of pancreatic tissue necrosis results in auto-digestion and local inflammation [3] and leads to sepsis, multiple organ injury, failure, and even death [4]. Oxidative/nitrosative stress and inflammation proved significant in acute pancreatitis (AP) pathogenesis [5] and multi-organ damage [6,7]. Pancreatic injury further increases the production of inflammatory cytokines and reactive oxygen/nitrogen species (ROS/RNS) [4]. As a part of multiple organ injury/failure, extra-pancreatic tissue damage associated with AP includes the circulatory system, kidneys, and liver [8]. The kidneys are among the most commonly affected organs in patients with AP, and acute kidney injury (AKI) has long been recognized as a common and severe complication of AP [9]. The essential inflammatory mediators involved in kidney tissue damage are nitric oxide (NO) and a large number of ROS. Therefore, it is assumed that the xanthine oxidase (XO) inhibitor, allopurinol, may prevent severe inflammation associated with this type of injury [8].

Oxygen free radicals have the potential to react with proteins and enzymes, leading to lipid peroxidation of cell and organelle membranes. This process can also result in protein denaturation, increased capillary permeability, ischemia, and direct injury to kidney cell membranes [10]. It is important to note that L-arginine has the capability to induce vascular vasodilation, potentially serving as a preventive measure against kidney injury in the aforementioned conditions. Furthermore, NO plays a role in renal injury by interacting with ROS and exerting a direct toxic effect on renal tubules. Additionally, NO reduces the responsiveness of blood vessels to stagnated substances, contributing to renal ischemia [11]. In AP, cytokines such as tumor necrosis factor-alpha (TNF-α) and interleukin-6 (IL-6) are released. The concentration of these cytokines steadily increases during the progression of AP and may play a contributory role in the pathogenesis of AKI [12].

Keeping these previous facts in mind, one could say that antioxidant and anti-inflammatory agents have a beneficial role in treating the mentioned disorder. A recently published study revealed that *Origanum vulgare* essential oil exerts antioxidant and anti-inflammatory activity and protects kidney tissue DNA damage from exposure to aflatoxin B1 [13]. The pharmacological properties of essential oils from other species, such as *O*. *majorana*, confirm its traditional uses. Thus, it was found that *O*. *majorana* essential oil possesses remarkable antimicrobial, antioxidant, anticancer, anti-inflammatory, antimutagenic, nephroprotective, and hepatoprotective activities [14]. The most significant components of the oregano essential oil are monoterpene phenols, carvacrol and thymol, and up to now, it was believed that their primary mode of action was the prevention of oxidative tissue damage through their antioxidant capacity [15]. More recently, a combination of thymol and carvacrol has been shown to have a synergistic protective effect in a model of cisplatin-induced nephrotoxicity, and the actions of these compounds might be attributed to their antioxidant, anti-inflammatory, and antiapoptotic activities [16]. Furthermore, treatment with thymol protects against the doxorubicin-induced toxic changes in kidney and heart function and histological integrity, which is mediated via the suppressive effect on oxidative stress and the enhancement effect on antioxidant defense systems [17]. It has already been demonstrated that allopurinol, *O*. *vulgare*, carvacrol, and thymol reduce the generation of ROS in the inflamed pancreas tissue and their consequential formation in distant organs [8,15].

This present study aimed to investigate the nephroprotective effects of *O*. *vulgare* essential oil and its main constituents, thymol and carvacrol, in rats injected with high doses of L-arginine. This was to be done by measuring serum and tissue parameters that reflect kidney functions and will include oxidative tissue damage parameters and parameters associated with tissue inflammation and cell death. Additionally, the histopathological analysis of kidney samples was conducted to evaluate micromorphological and immunological changes in tissue, hoping to better understand the cells/molecules involved in tissue damage.

## 2. Results

### 2.1. Essential Oil Analysis

Major essential oil constituents included carvacrol (44.6%), linalool (33.1%), thymol (2.5%), (E)-caryophyllene (1.9%), and other less abundant compounds (Appendix A). Constituents of the essential oil were mainly oxygenated monoterpenes (ca. 85%), while oxygenated sesquiterpenes were least abundant (ca. 0.5%). The analyzed essential oil constituents comprised 99.2% of the oil composition (Appendix A).

### 2.2. Serum Biochemical Analysis

This present study is designed as an acute model (24 h) of rat kidney damage that follows AP induced by a single intraperitoneal injection of L-arginine (3.5 g/kg of body weight). In the described model, the potential protective properties of acutely (1 h prior to AP) orally administered *O*. *vulgare* essential oil and its main constituents, thymol and carvacrol, were investigated. Appropriate control groups were treated with *O*. *vulgare* essential oil and its main constituents in the same manner as those in which AP was induced.

The results of this present study revealed that the mean serum activity of amylase, lipase, and lactate dehydrogenase (LDH) and the values of urea and creatinine were significantly elevated in the pancreatitis group compared to the vehicle-treated control group (*p* < 0.001) (Table 1). The application of *O. vulgare* essential oil (50 mg/kg), thymol (10 mg/kg), carvacrol (10 mg/kg), or the combination of thymol and carvacrol (1:1, *w/w*, 10 mg/kg) before L-arginine statistically significantly prevented an increase in the values of the previously mentioned serum parameters (*p* < 0.001) (Table 1). The pronounced activity of the combination of thymol and carvacrol, compared with the activity of the individual ones at the same dose, was detected in the urea and creatinine serum levels (Table 1). Further, our study showed that administering L-arginine produced a prominent rise in serum potassium levels when measured 24 h after the injection. At the same time, changes in the values of sodium were not detected (Table 1). The significant activity of the combination of thymol and carvacrol compared to the activity of the individual ones was clearly visible and manifested as decreased potassium serum levels (Table 1). It is worth mentioning that the acute administration of the test compounds on their own led to a slight elevation in the serum urea levels but had no impact on the creatinine or electrolyte levels (Table 1).

### 2.3. Tissue Oxidative Stress

The acute exposure of rats to test substances slightly increased tissue TBARS levels, while, at the same time, it had no impact on catalase activity (Table 2). On the other hand, in the kidney tissue of animals 24 h after L-arginine injection, a decrease in catalase activity was detected (Table 2). Tissue TBARS levels were still significantly higher in animals treated with L-arginine and test substances than in the vehicle-treated control (Table 2). It is worth mentioning that pre-treatments with *O. vulgare* essential oil and its main constituents prevented a decrease in catalase activity in comparison to L-arginine-treated rats (Table 2).

### 2.4. Kidney Tissue Damage Parameters

Specific tissue damage parameters, neutrophil-gelatinase-associated lipocalin (NGAL), and kidney injury molecule-1 (KIM-1) were found to be increased in kidney tissue 24 h following the application of L-arginine (*p* < 0.001) (Figure 1A,B). Essential oil of *O. vulgare* (50 mg/kg), thymol (10 mg/kg), carvacrol (10 mg/kg), or their combination (1:1, *w/w*, 10 mg/kg) applied prior to L-arginine injection significantly reduced the values of the mentioned parameters (*p* < 0.001) (Figure 1A,B). The values of KIM-1 in the kidney tissue of animals that did not receive L-arginine were below the detection limit, indicating that the cascade of events triggering KIM-1 formation was not initiated (Figure 1B). The same could be applied to the results from animals treated with essential oil and L-arginine, thymol and L-arginine, and allopurinol and L-arginine (Figure 1B).

### 2.5. Kidney Tissue Inflammatory Parameters

Inflammation-related parameters, such as NO, myeloperoxidase (MPO), and TNF-α, were found to be significantly increased, compared to the vehicle-treated control group, in the kidney tissue of rats 24 h after L-arginine application (Figure 2A–C). The administration of *O vulgare* essential oil and its main constituents significantly prevented an increase in NO concentration and MPO activity (Figure 2A,B) and, at the same time, did not significantly affect an increase in TNF-α concentration (Figure 2C). Treatment with allopurinol statistically significantly prevented an increase in all three of the tested inflammation-related parameters (Figure 2A–C). Application of the test compounds on their own did not change the concentration and/or activity of the tested parameters (Figure 2A–C).

### 2.6. Kidney Tissue Caspase-3 Activity

In the vehicle-treated control group animals, there was no measurable caspase-3 content (Figure 3), indicating no apoptosis in the kidney tissue of these animals. Interestingly, in animals treated with a single dose of 50 mg/kg of *O*. *vulgare* essential oil, a slight increase in caspase-3 content was detected (Figure 3). Animals treated with L-arginine had significantly increased caspase-3 content compared to the vehicle-treated control group (Figure 3). The most pronounced effect among the test compounds was that of the *O*. *vulgare* essential oil, whose application significantly prevented an increase in caspase-3 under the influence of L-arginine (Figure 3). Also, pre-treatment with thymol and carvacrol in combination prior to the administration of L-arginine exerted a significant impact on caspase-3 content elevation, which was comparable to the activity of allopurinol (Figure 3). Applied on their own prior to L-arginine, thymol and carvacrol produced a less pronounced effect than their combination (Figure 3).

### 2.7. Kidney Tissue Histopathological Analysis

Histopathological analysis of the kidney tissue included general analysis, which aimed to evaluate the extent of tissue damage and observe detectable micromorphological changes. Histochemical PAS staining was used to visualize tubular defects and the accumulation of casts within tubule lumens in the kidney tissue of animals with different treatment regiments. The expression of CD95 was used in order to confirm the activation of apoptotic signaling [18] within the kidney tissue. In contrast, CD68 immunostaining was used for the estimation of the presence of this molecule in tissue mononuclear cells [19].

Kidney tissue samples from vehicle-treated animals showed regular histomorphology, except for focal inflammation in one case (Figure 4A–E). Glomerular hypertrophy expressed as enlarged glomeruli and narrower Bowman’s space appeared in almost all experimental groups (Table 3). The scores differed from mild to significantly pronounced hypertrophy in animals with L-arginine without exposure to any other drug (Table 3, Figure 4F). Also, the same degree of cloudy swelling of the tubules was seen in all the treated animals, especially in a group with L-arginine and L-arginine and thymol treatment (Figure 4F,H). The analysis of the tissue samples from the majority of experimental groups did not show necrotic areas, except in traces in the group with L-arginine (Figure 4F) and animals exposed to L-arginine and thymol/allopurinol (Figure 4H,K). On the other hand, tubular hyaline casts were not observed in the control and the group which was only treated with *O*. *vulgare*, while other animals had occasional tubular deposits. In most of the treated animals, mild lymphocyte infiltration was detected; this was not visible in the group exposed to carvacrol L-arginine and allopurinol. Vascular congestion, minimal to a mild degree, was present in all the experimental groups but only absent in healthy, untreated animals (Figure 4).

PAS-stained kidney tissue samples obtained from the vehicle-treated animals (control group) and animals treated with *O. vulgare* and carvacrol did not show any PAS-positive tubular casts, except occasional staining localized on the proximal tubule brush border (Figure 4L–N). On the other hand, tubular cast visualized by PAS histochemical staining was focally seen in the groups treated with thymol and both carvacrol and thymol, and they were in the form of homogenous eosinophil/hyaline collection/mass/deposits in the tubular lumen (Figure 4O,P). PAS-positive tubular casts were focally seen in the L-arginine control group (Figure 4Q), and the group treated with allopurinol (Figure 4V) presented as a minor hyaline deposit on the luminal side of the proximal tubules. Also, moderate representation of the PAS-stained tubular collections was confirmed in the animals treated with L-arginine and thymol (Figure 4T) and in the group of animals treated with L-arginine and carvacrol and thymol (Figure 4U). On the other hand, applied histochemical staining did not visualize tubular casts in the group treated with L-arginine and either essential oil or carvacrol (Table 3).

The kidney tissue of animals in the vehicle-treated control and groups treated with test compounds was almost without any visible immunopositivity for CD95 protein (Figure 5A–E), except for occasional cells in the group that received *O. vulgare* essential oil (Figure 5B). The expression of CD95 followed a characteristic pattern revealing the most intense positivity in groups treated with L-arginine located on the surface of the cells belonging to the distal and collective ducts, rarely in the proximal tubule cells, and seldom in the glomerular structures (Figure 5F–K). The extent of this expression varied between the groups, but it was visible in all animals exposed to L-arginine after the test compounds (Figure 5F–K and Table 3).

Immunopositivity for CD68 was found to be slightly increased in the kidney tissue of rats belonging to the vehicle-treated control and groups treated with pure compounds (Figure 5L,N,O,P and Table 3), while in the group treated with *O. vulgare* essential oil, the positivity was slightly higher (Figure 5M). In animals that received L-arginine, a large number of cells positive for CD68 were noticed in the kidney interstitium either between the tubules or around the blood vessels (Figure 5Q–V). The most pronounced expression was in the group that received only L-arginine (Figure 5Q and Table 3), while the least pronounced expression occurred in the group that received L-arginine and allopurinol (Figure 5V and Table 3). The combination of the two test compounds had almost identical action as the compounds were applied on their own after L-arginine (Figure 5S–U and Table 3), while the essential oil had a slightly higher impact than the compounds (Figure 5R).

## 3. Discussion

Renal functions and tissue damage in experimental animals from different treatment groups were assessed by monitoring several biochemical parameters, serum amylase, lipase and LDH activities, urea, creatinine, sodium and potassium levels, and kidney tissue NGAL, KIM-1, MDA, and catalase. In addition to urea and creatinine, typically used to estimate kidney function, tissue levels of NGAL and KIM-1 were measured since both proteins are recognized as important markers of substantial renal tissue disorders [20,21]. Furthermore, tissue inflammation response was estimated based on kidney NO, MPO, and TNF-α levels, while apoptotic processes were determined based on caspase-3 content and immunohistochemical profile.

The results of the present study revealed no abnormality in the serum levels of the investigated kidney damage parameters in animals treated with only *O. vulgare* essential oil and its constituents, apart from a slight increase in urea levels (Table 1). Since the urea increase is independent of other kidney-damage-associated parameter levels (Figure 1A,B), this increase that follows the application of the essential oil and its constituents is not a consequence of direct kidney tissue damage. Namely, urea levels could be influenced by different external factors, which include protein intake, various metabolism-related states (e.g., increased catabolism), dehydration, etc. [22]. Thus, we can say that a potential limitation of this study is the serum urea and creatinine levels since they do not always depict a true image of kidney function.

Furthermore, the results demonstrated that the mean serum values of urea and creatinine were significantly elevated in rats in the L-arginine-treated group. After injecting a single high dose of L-arginine (3.5 g/kg), kidney injury potential follows the development of AP. The renal functions were impaired probably due to renal tubular system injury caused by either L-arginine and/or pancreatic enzymes, leading to increased serum creatinine levels [23]. This was visible during the histopathological analysis of the kidney tissue from rats in the group treated with L-arginine, which showed moderate glomerular hypertrophy and tubular degeneration in the form of cloudy swelling, followed by occasional tubular necrosis (Table 3). Also, the changes in serum urea and creatinine could be the consequence of a decrease in glomerular filtration, which again might be explained by the microscopic changes in glomerulus observed during pathohistological analysis (Table 3). These data suggest that either AP and/or a high dose of L-arginine cause mild AKI, which can be concluded based on the obtained biochemical and histopathological findings. Interestingly, treatment regiments caused a significant drop in serum creatinine levels, which could either be the consequence of increased glomerular filtration or increased tubular secretion.

Also, administering cationic amino acid L-arginine produced a prominent increase in serum potassium levels when measured 24 h after the injection. At the same time, changes in sodium values were not detected. A recently published study also revealed that the systemic (*i.p.*) application of L-arginine produced a marked rise in plasma potassium levels in mice [24]. Potassium is required for vital cellular function, and even small changes in the distribution between intra- and extracellular compartments can result in a marked elevation in serum potassium values [25]. Disturbance in serum potassium levels, either an increase or decrease, is almost certainly the consequence of kidney disorder or related to a certain degree to kidney dysfunction. Different studies described that the efflux of potassium from cells to the extracellular spaces was associated with administering L-arginine and other cationic amino acids [24]. The results from the present study emphasized the significant effect of thymol and carvacrol applied in combination on a decrease in the potassium serum level (Table 1), contributing to the fine regulation of cation homeostasis.

One could argue that the changes in kidney tissue and consequential changes in the studied serum parameters might not only be related to enzymes and damage products coming from the affected pancreatic tissue. Our previous publication showed that L-arginine-induced AP is followed by a milliard of events in the peritoneal cavity, which include fluid and cell extravasation, enzyme leakage, and inflammation activation [6]. This kind of fluid extravasation to the peritoneal cavity (so-called third spacing) most probably contributed to the observed AKI by reducing the glomerular filtration rate.

In the model of L-arginine-induced AKI, *O. vulagare* essential oil, thymol, and carvacrol were shown to prevent kidney oxidative lipid damage estimated through the measured TBARS levels (Table 1). The amount of TBARS is considered to reflect cell and/or organelle membrane ROS-mediated damage [26]. Considering the fact that essential oil and the tested monoterpenes are potent antioxidants [27], it is not surprising that they prevented an increase in TBARS levels. Treatment with allopurinol, a specific XO inhibitor, also prevented an increase in TBARS concentration induced by L-arginine, suggesting the beneficial properties of drugs that inhibit XO, as suggested elsewhere [2,6,7,8].

An increase in the catalase activity in animals treated with L-arginine and/or *O. vulgare* essential oil, and its main constituents, indicates that the test compounds could play an important role in scavenging hydrogen peroxide and modifying catalase expression/activity (Table 2). Namely, catalase is an enzyme that directly catalyzes the transformation of peroxides and superoxides to non-toxic oxygen species [28]. In our study, the potential explanation for a marked decrease in kidney tissue activity after L-arginine application could be the consequence of treatment with a high dose of L-arginine, on the one hand, and elevated citrulline metabolism on the other. The action of citrulline on catalase activity, as well as on some other antioxidant enzymes, arises from the results of a previous study showing the inhibitory action of citrullinemia and hyperammonemia [29]. This is highly plausible in the present state since the urea levels in these groups are significantly increased (Table 1), and the citrulline itself, as a product of L-arginine degradation, must also be increased and follow the observed NO increase (Figure 2). Further studies could be undertaken to elucidate the mechanisms involved in enhancing catalase activity and the protective potential of the test compounds, which could be numerous and synergistic.

The upregulation of KIM-1 is caused by the dedifferentiation of cells of the proximal tubules [21] and during the states where the damaged kidney cells are trying to remove cell debris [30]. Having in mind that the production of ROS is the foundation of AP progression and high L-arginine dose effects, in our investigation, we evaluated KIM-1 kidney tissue levels in order to potentially find a connection between AP and kidney tubule cell damage (Figure 1B). KIM-1 is known to be activated potentially via mitogen-activated protein kinase and by the disturbance of kidney oxidative capacities [30]. Thus, it could be brought in connection with the inflammatory processes in kidney tissue after applying L-arginine (Figure 2 and Figure 5Q–V). The obtained result related to thymol activity in preventing a rise in the KIM-1 level (Figure 1B) is in accordance with the findings of a previous study [31]. On the other hand, carvacrol moderately influenced KIM-1 kidney levels (Figure 1B), which has also been proven recently, although in a five-fold higher dose and during a seven-day experiment [32]. Based on these results, the activity of *O*. *vulgare* essential oil could be mainly attributed to the activity of thymol activity, and to a lesser percentage to that of carvacrol. However, since this oil mainly contains carvacrol (Appendix A), other mechanisms might be underlining the activity of the oil other than the proposed one associated with thymol. Also, it should be noted that no significant synergistic potential on KIM-1 levels was observed in the group that received both thymol and carvacrol.

Another kidney damage marker, NGAL, is known to be upregulated in kidney tubules within hours after harmful stimulus, indicating the possibility that this protein belongs to that limited panel of stress-induced renal biomarkers involved in the pathophysiological process of AKI [33]. For example, NGAL is rapidly produced and released from tubular cells in mice after the intraperitoneal injection of high doses of cisplatin, a well-known agent that causes tubular necrosis [34]. Our study revealed that KIM-1 and NGAL were increased following the application of L-arginine (Figure 1A,B). The application of either thymol, carvacrol, or their combination prior to L-arginine injection significantly reduced the values of the previously mentioned kidney damage parameters (Figure 1). These results could be related to the direct ROS scavenging ability of the mentioned compounds [27] and to some mechanisms that prevent ROS generation or increase in their removal.

Inflammation-related parameters, such as NO, MPO, and TNF-alpha, were found to be significantly increased in kidney tissue after L-arginine application (Figure 2A–C). The literature data confirmed that a high amount of MPO is a significant sign of intensive phagocyte cell accumulation in the peritoneal cavity, i.e., neutrophils and macrophages. Under intensive inflammation, such as that seen after L-arginine injection and which could be related to AP, the peritoneal cavity field with MPO-releasing cells becomes a place of extensive ROS production and a site where further signaling/chemotaxis occurs [35]. On the other hand, NO mediates kidney injury by interacting with ROS production and produces a direct toxic effect on renal tubule cells [11]. Also, NO and L-arginine could partially play a protective role in these situations since they are known to be vasodilators. Finally, TNF-α acts directly on pancreatic duct cells, glomerular cells, and renal tubule cells, causing ischemia and necrosis and contributing to inflammation. Excessive TNF-α enters the circulation and further stimulates neutrophils, thus promoting their aggregation and aggravation of target tissue injury [36].

The administration of *O*. *vulgare* essential oil and its main constituents after nine applications of L-arginine prevented an increase in NO concentration and MPO activity to a certain extent (Figure 2A,B). Interestingly, no impact on the elevation in TNF-α concentrations was noted (Figure 2C), suggesting that the anti-inflammatory activity of the tested compounds arises from the modulation of inflammatory mediators’ production and fine regulation of pro-inflammatory cytokine synthesis in the kidney. The production of these molecules is associated in part with CD68-positive cells and macrophages, whose number has increased after L-arginine application (Figure 5Q). Also, treatment with allopurinol prevented an increase in all three tested inflammation-related parameters induced by L-arginine (Figure 2 and Figure 5V). Recently published findings have demonstrated that carvacrol suppresses the expression of inflammatory marker genes such as TNF-α, IL-6, inducible NO synthase (iNOS), cyclooxygenase 2, and nuclear factor kappa-B in D-galactosamine hepatotoxic rats [37], explaining the impact on NO concentration, but not on TNF-α. Also, the observed activity towards the production of NO exerted by thymol can be explained through the inhibition of iNOS, which was proven in vitro in stimulated murine macrophages [38]. A slight diminution in the MPO activity can partially be attributed to the impact of thymol on the neutrophil adhesion capacity, which is decreased by exposure to this monoterpene [39]. Furthermore, the anti-inflammatory activity of thymol could be associated with NF-kB and PI3K/Akt signaling pathway attenuation, which was recently discovered in a glycerol-induced AKI model [40]. Interestingly, scarce data are found on the influence of thymol on TNF-α production; however, some of them do suggest the potential of this monoterpene to decrease its production/expression [41]. These data, in part, correspond with the extent of the inflammatory response (Table 3) and CD68 expression (Figure 5). The data observed cannot give complete insight into the activity of either oil or the combination of two monoterpenes, but rather they give an introduction and basis for future studies. A recent study showed that the combination of thymol and carvacrol protects kidney tissue from radiation-induced damage [42] by interacting with protein structures such as TNF-α, NF-kB insulin-like growth factor-1 (IGF-1), and calcitonin gene-related peptide.

The activation of CD95 (Fas ligand-receptor) located on the cell membrane is bottom-line associated with caspase-3 activation and cell death, contributing to acute kidney injury [43]. The expression patterns in the present experimental design revealed that CD95 in rats treated with L-arginine is mainly expressed in distal tubules and, to a lesser extent, in proximal tubule cells (Figure 5F–K). Increased NO formation in groups treated with L-arginine (Figure 2A) led to the activation of CD95 and further pushed the cells into degenerative processes, as is seen in T lymphocytes [44]. The findings on the NO content in rats treated with L-arginine and some of the test substances should indicate that there should not be a significant number of CD95-positive cells in kidney tissue, which is not the case here (Figure 5).

Another potential reason that impacts the sensitivity of tubule cells to L-arginine is their role in its elimination from the urine. Namely, proximal tubule cells’ function is related to the absorption of amino acids via several transporters, while the same does not occur in distal and convoluted tubule cells [45]. Thus, when the transport maximum is reached, the remaining L-arginine flows down to distal tubules where the urine is being concentrated/diluted. These urine concentration changes probably increase the L-arginine concentration above the values when it becomes toxic to tubule system cells. Following this hypothesis, the results of the expression pattern of CD95 in rats exposed to L-arginine could be better understood.

Moreover, activated monocytes, recognized through the expression of CD68, are known to secrete CD95 ligand, which activates CD95 receptors [46]. The CD95 ligand (called FasL) is a type II transmembrane protein from the TNF family of death factors, which, after its contact with the CD95 receptor, triggers the cascade, leading to the activation of caspase-8 [47]. Thus, an increased expression of CD86 could be correlated well with an increased CD95 expression, pointing to the importance of activated monocytes in the pathophysiology of kidney tissue damage. Also, tested oregano essential oil and its components (thymol and carvacrol) influence monocyte/macrophage activation in vitro and in vivo [6,48,49]. By doing so, the tested oil and active constituents may dampen the damage induced by the secretion of CD95 ligand and the activation of CD95, which is evident from the obtained results (Figure 5R–U).

Activated caspase-3 is considered a central apoptotic effector enzyme located beyond the point of no return in the cell death program. Caspase-3 activation takes place in the effector phase of the apoptotic process, and its detection, therefore, represents a reliable tool for the identification of apoptotic cells [50] since cells with the prominent activation of caspase-3 are surely going to die [51,52]. Initiation of the caspase cascade finally leads to the cleavage of proteins critical for cell survival and the activation of endonucleases, resulting in DNA fragmentation [52]. The results of the present study revealed that animals exposed to a high dose of L-arginine had increased caspase-3 content, while pre-treatment with *O. vulgare* essential oil or combination with thymol and carvacrol significantly prevented an increase in this parameter under the influence of L-arginine (Figure 3). The obtained results related to the carvacrol activity are in accordance with the previously reported data, showing that it exerts an extensive anti-apoptotic property through the reduction in the caspase-3 activity [53,54]. During a two-week experiment, the application of thymol in a dose of 20 mg/kg to rats prevented a kidney tissue caspase-3 increase induced by the application of gentamicin, a well-known nephrotoxic antibiotic [31], potentially explaining the data related to the thymol activity. Thus, the observed activity of a combination of thymol and carvacrol almost corresponds to the total activity of the *O*. *vulgare* essential oil, and we could potentially say that the two compounds synergistically act in preventing caspase-3 increase.

Oregano oil monoterpene phenols, thymol, and carvacrol were previously proven to possess a large number of pharmacological activities, including nephroprotective activity [31,40,42,54]. In accordance with previous findings, the results of our study demonstrated that thymol and carvacrol alone or combined significantly reduced serum urea and creatinine levels, as well as kidney tissue NGAL, KIM-1, and inflammatory parameters (NO, MPO, and TNF-α) and decreased CD68 expressing cells infiltration, thus preventing renal damage caused by a high dose of L-arginine. Considering that there are already reports about the synergistic action between thymol and carvacrol in antioxidant assays [27], the present study’s findings emphasized the more significant nephroprotective potential of the combination of the two compounds, which could be attributed to synergism. The mechanisms studied here reflect the essential excretory kidney function and, to a certain extent, the inflammatory-based mechanism underlying the present model of kidney damage. Also, one should bear in mind that damage to kidney tissue is, to a significant portion, lessened by the diminution in pancreatic damage which arises from the oil and compound application.

## 4. Materials and Methods

### 4.1. Chemicals and Reagents

Thymol (T), carvacrol (C), and L-arginine were obtained from Sigma-Aldrich (St. Louis, MO, USA), while the specific xanthine oxidase inhibitor, allopurinol, was acquired from Alfa Aesar (Kandel, Germany). *Origanum vulgare* essential oil was obtained from commercial sources (Siempre Viva Oils, Serbia). All compounds (OR, C, and T) were dissolved in olive oil and given to animals orally in a volume not exceeding 0.2 mL. GC and GC-MS analyses of the essential oil, under standard conditions [55], were carried out using a Hewlett-Packard 6890N gas chromatograph equipped with a fused silica capillary column Optima 5MS (5%-diphenyl-) 95%-dimethylpolysiloxane, 30 m × 0.25 mm, film thickness 0.25 μm, and coupled with a 5975B mass selective detector (Agilent Technologies, Santa Clara, CA, USA). Linear retention indices were calculated for all the identified components using standards of n-alkanes (C7–C30). The chemical composition of the essential oil is presented in Appendix A Appendix A.

### 4.2. Animals

Adult (four months old) male Wistar rats weighing 300–350 g were kept under standard husbandry conditions, which included room temperature of 23 ± 2 °C, air humidity of 60%, and 12/12 (light/dark) cycle, with standard laboratory chow (Vivarium of the Institute of Biomedical Research, Faculty of Medicine, University of Nis, Serbia) and water being provided to the animals without restriction (ad libitum) during the experiments. The research procedures adhered to the principles outlined in the Declaration of Helsinki and the European Community guidelines for the ethical treatment of laboratory animals, specifically in accordance with the EU Directive of 2010 (2010/63/EU). The experimental protocols were initiated after receiving approval from the animal Ethics Committee, as indicated by decision number 323-07-00278/2017-05/2. All necessary measures were taken to minimize animal suffering, and efforts were made to limit the number of animals used in the study.

### 4.3. Experimental Design

L-arginine in a dose of 3.5 g/kg, dissolved in saline buffer, was administrated via an intraperitoneal (*i.p.*) route to rats [6,7] 1 h after the per os (*p.o.*) application of the test sample/compounds. All (n = 66) animals were randomly divided into eleven experimental groups, with six animals per group. Each group was caged together one week prior to and during the experiment. The treatment of animals was as follows:Group I (OO group/control group): animals were given olive oil *p.o.* in a 10 mL/kg dose.Group II (OR): animals were given *O. vulgare* essential oil (OR) *p.o.* in a 50 mg/kg dose.Group III (C): animals were given C *p.o.* in a 10 mg/kg dose.Group IV (T): animals were given T *p.o.* in a 10 mg/kg dose.Group V (CT): animals were given a combination of C and T *p.o.* (1:1, *w/w*, 10 mg/kg dose).Group VI (AOO): animals were given olive oil *p.o.* in a 10 mL/kg dose + a single dose of L-arginine.Group VII (AOR): animals were given OR *p.o.* in a 50 mg/kg dose + a single dose of L-arginine.Group VIII (AC): animals were given C *p.o.* in a 10 mg/kg dose + a single dose of L-arginine.Group IX (AT): animals were given T *p.o.* in a 10 mg/kg dose + a single dose of L-arginine.Group X (ACT): animals were given a combination of C and T *p.o.* (1:1, *w/w*, 10 mg/kg) + a single dose of L-arginine.Group XI (AA): animals were given allopurinol (A) *p.o.* in a 100 mg/kg dose + a single dose of L-arginine.

Twenty-four hours after the administration of L-arginine, animals were sacrificed by an overdose of ketamine (Ketamidor 10%, Richter Pharma AG, Wels, Austria). Blood samples were withdrawn by a cardiac puncture to evaluate the biochemical parameters and were kept at −80 °C until use. Kidneys were removed, and samples were separated for biochemical and histological analyses.

### 4.4. Serum Biochemical Measurements

Blood was centrifuged at 1500 rpm at 4 °C for 15 min to obtain the serum in which urea and creatinine, potassium, and sodium levels were assayed using the Olympus AU680 Chemistry-Immuno Analyzer (Beckman Coulter, Barcelona, Spain). Furthermore, the values of markers reflecting pancreatic damage, i.e., activities of enzymes alpha-amylase, lipase, and lactate dehydrogenase (LDH), were also determined.

### 4.5. Tissue Isolation and Homogenate Preparation

Dissected pieces of kidney tissue were homogenized in a 10-times higher volume (10%, *w/v*) of phosphate-buffered saline and centrifuged at 12,000 rpm at 4 °C for 20 min. Tissue parameters were determined using clear supernatants obtained after centrifugation. The protein content in the supernatants was measured and determined by Lowry’s method [56] using the standard curve of bovine serum albumin.

### 4.6. Tissue Inflammatory Parameters

#### 4.6.1. Nitric Oxide (NO) Assay

The concentration of nitrites (NO_2_^−^) in the supernatants, as an indicator of NO production, was determined by the Griess reaction. Briefly, 50 μL of supernatants was transferred to an empty microplate. The prepared Griess reagent (50 μL) was added to each well, and the incubation continued in the dark for another 10 min. Finally, the absorbance of each well was measured at 540 nm using a microplate reader. The concentrations of nitrites were calculated from the nitrite standard curve equation and expressed in μM.

#### 4.6.2. Myeloperoxidase (MPO) Activity Determination

The activity of MPO was determined using *o*-phenylenediamine (1,2-diaminobenzene) as the color reagent, and the enzymatic reaction was initiated by adding H_2_O_2_ [6]. The reaction was stopped with an aqueous H_2_SO_4_ solution, and the formed product’s optical densities (ODs) were determined at 540 nm. The results are expressed as ODs (absorbance at 540 nm) × 1000.

#### 4.6.3. Production of TNF-α Cytokine—ELISA Assay

Supernatants of kidney homogenates were collected and subjected to the quantification of TNF-α pro-inflammatory cytokine production. The quantification was performed using enzyme-linked immunosorbent assay (ELISA) kits according to the manufacturer’s suggestions (R&D Systems, Minneapolis, MN, USA). The results are expressed in pg/mL.

### 4.7. Tissue Damage Parameters

Collected supernatants were kept at −80 °C until their usage for the determination of neutrophil-gelatinase-associated lipocalin (NGAL, Rat Lipocalin-2 ELISA kit, Abcam (Cambridge, UK), ab207925), kidney injury molecule-1 (KIM-1, Rat TIM1/KIM-1/HAVCR Quantikine^®^ ELISA, R&D SYSTEMS^®^, RKM100), and caspase-3 content (Rat caspase-3 (Ser-29) ELISA Abcam, ab181418). Assays were conducted following the manufacturer’s instructions.

#### 4.7.1. Kidney TBARS Levels Determination

The quantification of thiobarbituric acid reactive substances (TBARSs) was conducted using a standard method, as previously outlined [6]. In brief, 100 μL of the kidney tissue homogenate was subjected to boiling at 95 °C in the presence of a thiobarbituric acid solution until the development of chromogenic color. The intensity of the resulting-colored reaction product was then measured at 540 nm using a Multiscan Ascent spectrophotometer (Labsystems, Vantaa, Finland). The amount of malondialdehyde (MDA) in each sample was determined by reference to a standard curve established using 1,1,3,3-tetraethoxypropane as the TBARS equivalent. The concentration of TBARS in the kidney tissues was expressed as nanomoles per milligram of kidney tissue proteins.

#### 4.7.2. Catalase Activity Determination

Catalase activity was measured spectrophotometrically after a reaction with H_2_O_2_ as a substrate and ammonium molybdate at 405 nm [57]. The activity of catalase was expressed as U/g of kidney tissue proteins.

### 4.8. Histopathological and Immunohistochemical Analysis

For histopathological examination, kidney tissue specimens were isolated and fixed in a buffered formaldehyde solution (10%, *w/v*). Subsequently, the fixed tissues underwent dehydration using ethanol solutions at varying concentrations (50–100%, *v/v*). Following dehydration, the tissues were embedded in paraffin and sliced into sections measuring 4–5 µm in thickness. These sections were then stained using hematoxylin and eosin (HE) as well as periodic acid-Schiff (PAS) staining techniques. Immunohistochemical staining was performed using primary monoclonal antibodies: CD68 (dilution 1:100, GeneTex, Irvine, CA, USA) and CD95 (dilution 1:10, Abcam). The standard procedure of antigen retrieval (citrate buffer) and endogenous peroxidase blockage (using 3% hydrogen peroxide) was performed before overnight incubation with the primary antibody in a moist chamber. The visualization was effectuated using diaminobenzidine and counterstained with Mayer’s hematoxylin.

Stained tissue sections were subjected to additional examination using an Olympus BH2 light microscope (Olympus America Inc., Valley, PA, USA) by two independent researchers (V.P. and M.S.) who were blinded to the nature of the treatment. The objective was to assess and quantify the extent of tissue damage. The evaluation of kidney morphological changes, based on hematoxylin and eosin (H&E)-stained tissue, involved a grading system: 0 (no change), 1 (mild changes; <30%), 2 (moderate changes; 30–50%), and 3 (severe changes; >50%). Kidney tissue staining with the PAS method was used for the visualization of tubular casts present in the lumen of kidney tubules. Immunohistochemical staining was graded as: absent/no staining (0), trace (0.5), mild (1), moderate (2), strong positive staining (3) [7].

### 4.9. Statistical Analysis

The results were presented as the mean ± standard deviation (SD). Statistical analyses to determine significant differences were conducted using a one-way analysis of variance (ANOVA), followed by Tukey’s post hoc test for multiple comparisons. The statistical analysis was performed using GraphPad Prism version 5.03 (San Diego, CA, USA). Probability values (*p*) less than 0.05 were considered statistically significant.

## 5. Conclusions

In summary, oregano monoterpene phenols thymol and carvacrol can prevent L-arginine-induced tissue damage and potentially improve the prognosis of pancreatitis by reducing damage to distant organs. In that sense, they can be used as potential agents for protecting/preventing kidney function impairment caused by high doses of L-arginine in rats. The damaged kidney tissue could be either from the high dose of L-arginine or from the pancreatic enzymes excreted, which alone or combined provoke oxidative tissue damage, inflammation, and cell apoptosis. The conclusions regarding the nephroprotective action of the test compounds were drawn from data evaluating the applied test compounds’ antioxidant, anti-inflammatory, and antiapoptotic effects and, above all, due to their ability to lessen the disturbances arising from acute pancreatic damage.

## Figures and Tables

**Figure 1 ijms-25-00941-f001:**
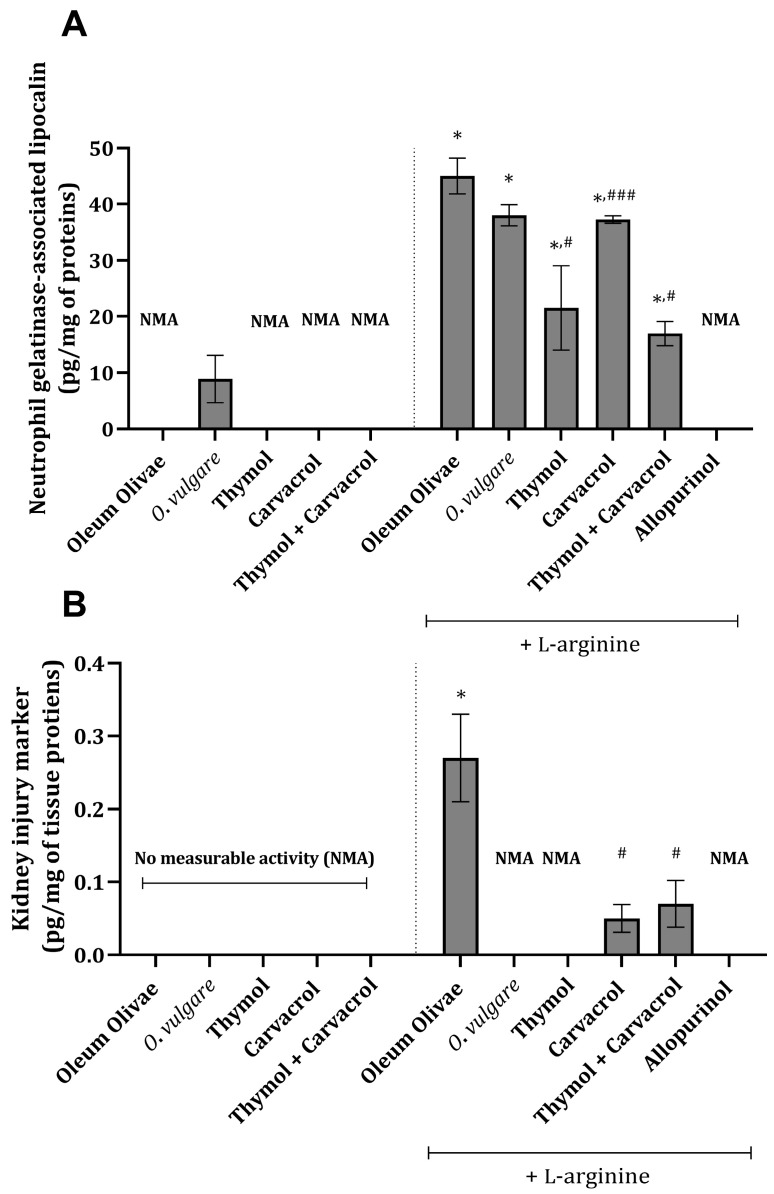
Kidney tissue NGAL (**A**) and KIM-1 (**B**) levels in rats belonging to different experimental groups. * *p* < 0.001 vs. vehicle-treated control (OO), ### *p* < 0.05; # *p* < 0.001 vs. AOO group; NMA—no measurable activity.

**Figure 2 ijms-25-00941-f002:**
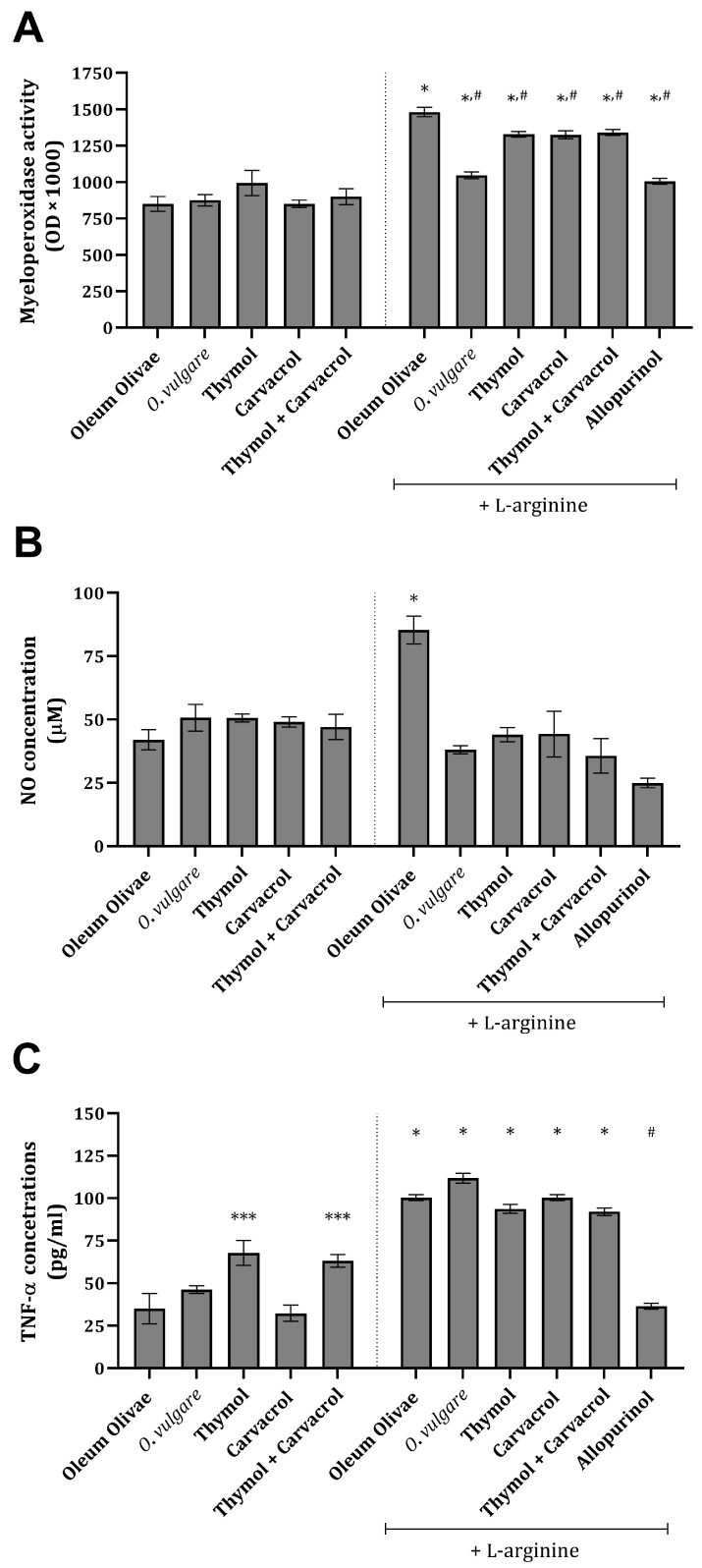
Kidney tissue MPO activity (**A**), NO (**B**), and TNF-α concentrations (**C**) in rats belonging to different experimental groups. *** *p* < 0.05; * *p* < 0.001 vs. vehicle-treated control (OO), ^#^
*p* < 0.001 vs. AOO group.

**Figure 3 ijms-25-00941-f003:**
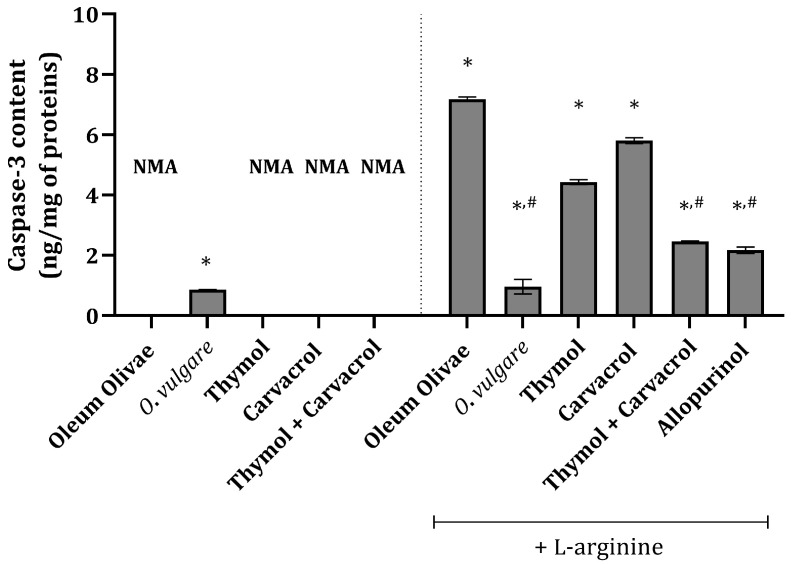
Kidney tissue caspase-3 content in rats belonging to different experimental groups. * *p* < 0.001 vs. vehicle-treated group (OO), # *p* < 0.001 vs. control; NMA—no measurable activity.

**Figure 4 ijms-25-00941-f004:**
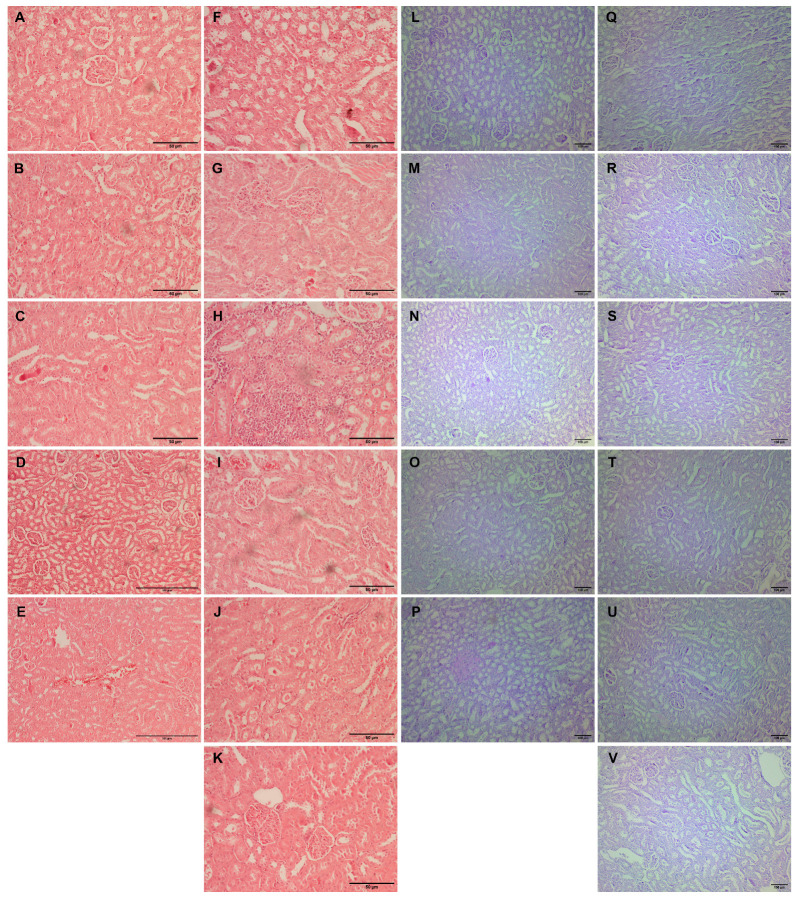
Histopathological analysis of rat kidney tissue stained with H&E (**A**–**K**, magnification ×200, except for **D**,**E,** where magnification is ×100) and PAS (**L**–**V**, magnification ×100). Rats belonging to the vehicle-, O. vulgare-, thymol-, carvacrol-, thymol, and carvacrol-treated groups (**A**–**E**,**L**–**P**, respectively) showed mainly regular kidney tissue structures with occasional inflammatory cell or tubular deposits. (**F**,**Q**) Kidneys of animals treated with L-arginine showed vascular congestion, tubular degeneration, inflammatory infiltration, occasional cell necrosis, and tubular deposits; (**G**,**R**) L-arginine and essential oil-treated animals showed minimal changes in the kidneys in the form of degeneration and occasional tubular deposit; (**H**,**S**) L-arginine and carvacrol-treated animals exerted pronounced changes in the form of tubular degeneration, inflammatory infiltrate, and blood stasis; (**I**,**T**) in L-arginine and thymol-treated animals, there was significant tubular damage and blood stasis; (**J**,**U**) the most prominent feature in L-arginine and carvacrol + thymol-treated animals was blood stasis and glomerular space narrowing; (**K**,**V**) in L-arginine and allopurinol-treated rats, only minimal changes in tubular cells and occasional glomerular space narrowing was noted.

**Figure 5 ijms-25-00941-f005:**
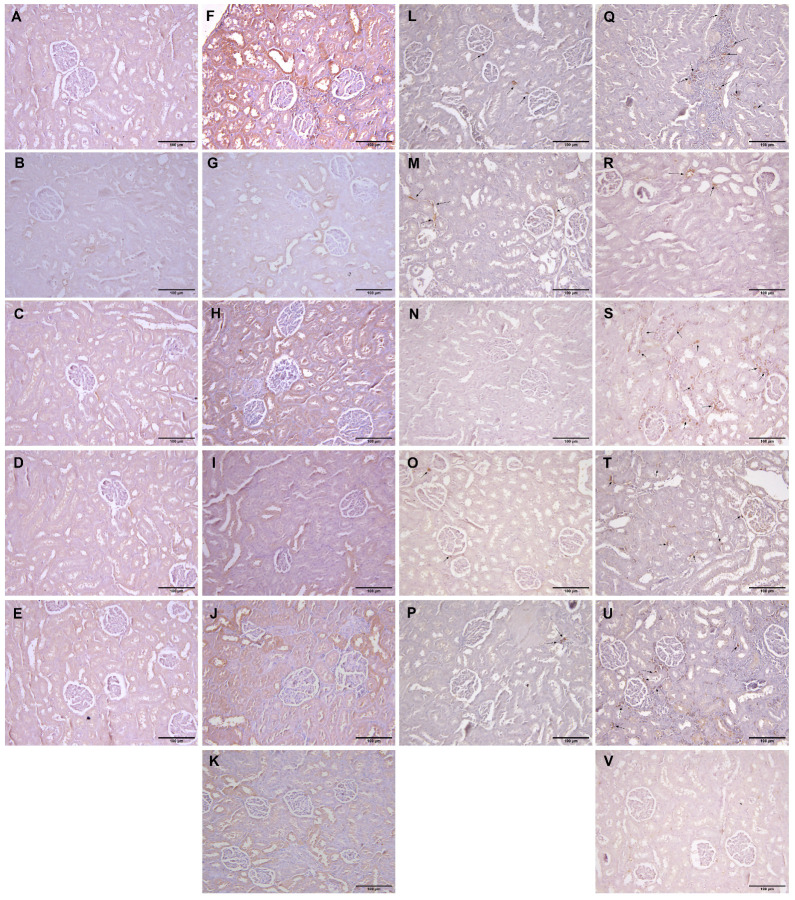
Histopathological analysis of rat kidney tissue stained with CD95 (**A**–**K**, magnification ×100) and CD68 (**L**–**V**, magnification ×100). Rats belonging to the vehicle-, *O*. *vulgare*-, thymol-, carvacrol-, thymol-, and carvacrol-treated groups (**A**–**E**,**L**–**P**, respectively) showed mainly regular kidney tissue structures with occasional weak immunopositivity in either kidney tubule cells (CD95) or resident inflammatory cells (CD68). (**F**,**Q**) Kidneys of animals treated with L-arginine showed significant CD95 expression in tubular cells and CD68 in inflammatory cells; (**G**,**R**) L-arginine and essential oil-treated animals showed moderate CD95 expression and rare CD68; (**H**,**S**) L-arginine and carvacrol-treated animals exerted mild tubular CD95 and extensive inflammation characterized by an increase in CD68 expression; (**I**,**T**) in L-arginine and thymol-treated animals, there was a significant tubular CD95 and mild inflammatory cell CD68 expression; (**J**,**U**) in L-arginine and carvacrol + thymol-treated animals, CD95 expression was mild, as was the expression of CD68; (**K**,**V**) in L-arginine and allopurinol-treated rats, only occasional CD95 expression was found, and almost no CD68-positive cells were visible. Arrows denote CD68-positive cells.

**Table 1 ijms-25-00941-t001:** The values of biochemical parameters in rat serum and kidney obtained from different experimental groups.

Group/Biochemical Parameter	Alpha-Amylase(U/L)	Lipase (U/L)	LDH (U/L)	Urea (mmol/L)	Creatinine (µmol/L)	Na (mmol/L)	K (mmol/L)
Vehicle-treated control (OO)	2155.6 ± 221	10.23 ± 2.8	1183 ± 108	6.3 ± 0.5	40 ± 1.5	140 ± 1.8	4.5 ± 0.1
*O. vulgare* essential oil (OR)	2247 ± 108.2	4.55 ± 0.9 *	1092 ± 195	7.42 ± 0.61 ***	41.28 ± 3.17	140.5 ± 1.91	4.7 ± 0.42
Thymol (T)	1984.3 ± 31.6	3.86 ± 0.2 **	1013 ± 19	7.7 ± 0.64 ***	40.8 ± 1.86	140.5 ± 2.52	4.83 ± 0.2
Carvacrol (C)	1708.6 ± 184.6 *	4.13 ± 0.6 **	1071.3 ± 31.1	7.15 ± 0.24 ***	41.4 ± 3.14	141.25 ± 2.87	4.77 ± 0.21
Carvacrol + thymol (10 mg/kg)	1658 ± 170.3 *	4.96 ± 1.2 *	1109 ± 42.3	7.22 ± 0.17 ***	41.1 ± 2.2	140.7 ± 2.14	4.6 ± 0.11
L-arginine + vehicle (AOO)	4453.2 ± 328 **	83.3 ± 13 ***	22,413 ± 831 ***	9.97 ± 0.42 **	48.5 ± 1.63 ***	143.3 ± 2.4	6.5 ± 0.5 ^#^
L-arginine + *O. vulgare* essential oil (AOR)	2338 ± 163.7 ^#^	6.7 ± 1.1 ^#^	1695.7 ± 208.6 ^#^	8.08 ± 0.97	43.3 ± 2.7	146.5 ± 1	5.33 ± 0.6
L-arginine + thymol (AT)	2203 ± 198 ^#^	4.9 ± 0.7 ^#^	1806.7 ± 217.2 ^#^	8.83 ± 1.57	15.77 ± 1.4 ^###^	138.3 ± 1.15	5.23 ± 0.16
L-arginine + carvacrol (AC)	2297 ± 151 ^#^	6.1 ± 0.95 ^#^	731.3 ± 119.5 ^#^	7.33 ± 0.99	22.1 ± 4.94 ***	147 ± 2	5.23 ± 0.93
L-arginine + thymol and carvacrol (ATC)	2511.8 ± 137 ^#^	18.76 ± 1.64 ^#^	831.8 ± 125.2 ^#^	6.5 ± 1.18 ^##^	11 ± 5.7 ***	140.6 ± 6.6	4.43 ± 0.1 *
L-arginine + allopurinol (AA)	2059 ± 124 ^#^	13.9 ± 1.1 ^#^	1571 ± 197.8 ^#^	6.97 ± 1.32 ^###^	24.77 ± 13.93 **	143.5 ± 2.65	5.25 ± 1.6

Data are given as mean ± SD and further compared with one-way ANOVA followed by Tukey’s post hoc test; * *p* < 0.001, ** *p* < 0.01, *** *p* < 0.05 vs. vehicle-treated control and **^#^** *p* < 0.001, **^##^** *p* < 0.01, **^###^** *p* < 0.05 vs. POO group.

**Table 2 ijms-25-00941-t002:** Oxidative-stress-related parameters in kidney tissue obtained from rats belonging to different experimental groups.

Group/Biochemical Parameter	TBARS (nmol/mg of Proteins)	Catalase (U/g of Proteins)
Vehicle-treated control (OO)	4.2 ± 0.49	314.3 ± 38.8
*O. vulgare* essential oil (OR)	6.9 ± 0.2 **	213.6 ± 30.9
Thymol (T)	6.6 ± 0.3 **	369 ± 74.2
Carvacrol (C)	6.4 ± 0.1 **	358.7 ± 60.4
Carvacrol + thymol (10 mg/kg)	5.8 ± 0.3 **	320.1 ± 65.4
L-arginine + vehicle (POO)	10.5 ±0.55 *	130.7 ± 28.8 *
L-arginine + *O. vulgare* essential oil (POR)	8.1 ± 0.23 ^##^	378 ± 85.3 *
L-arginine + thymol (PT)	7.9 ± 0.18 ^##^	247 ± 65.5 ^#^
L-arginine + carvacrol (PC)	8.4 ± 1.59 ^##^	215 ± 33
L-arginine + thymol and carvacrol (PTC)	8.6 ± 0.39 ^##^	178.7 ± 16
L-arginine + allopurinol (PA)	8.0 ± 0.23 ^##^	241 ± 39

Data are given as mean ± SD and further compared with one-way ANOVA followed by Tukey’s post hoc test; * *p* < 0.001, ** *p* < 0.01 vs. vehicle-treated control and **^#^** *p* < 0.001, **^##^** *p* < 0.01 vs. POO group.

**Table 3 ijms-25-00941-t003:** Histopathological and immunohistochemical scores obtained for different experimental groups.

Group/Histopathological Parameter	Glomerular Hypertrophy	Tubular Changes				Inflammation	Blood Stasis	CD95	CD68
		Cloudy Swelling	Tubular Necrosis	Tubular Cast	PAS-Positive Tubular Cast				
Vehicle-treated control (OO)	0	0	0	0	0	0.125	0	2	0.5
*O. vulgare* (50 mg/kg)	1	1	0	0	0	0.5	1	1	0.5
Carvacrol (10 mg/kg)	0.75	1	0	0.25	0	0	0.5	2	0
Thymol (10 mg/kg)	0.5	0.25	0	0.25	0.25	0.25	0.5	2	0.5
Carvacrol + thymol (10 mg/kg)	0.75	0.5	0	0.25	0.125	0.25	0.5	2	0.5
L-arginine + OO	2	2	0.5	0.75	0.25	2	1.75	3	0.5
L-arginine + *O. vulgare* (50 mg/kg)	0.75	1.25	0	0.25	0	0.5	1	1	0.5
L-arginine + carvacrol (10 mg/kg)	1	1.25	0	1	0	1.5	1	2	1
L-arginine + thymol (10 mg/kg)	1	2	0.5	1	0.75	1	2	1	0.5
L-arginine + carvacrol + thymol (10 mg/kg)	1.2	1.25	0	1	1	1	1.25	2	0.5
L-arginine + allopurinol (50 mg/kg)	1	1	0.25	0.25	0.25	0	1	1	0

0 (no change), 1 (mild changes; <30%), 2 (moderate; 30–50%), and 3 (severe changes; >50%). Immunohistochemical staining graded as: absent/no staining (0), trace (0.5), mild (1), moderate (2), strong positive staining (3).

## Data Availability

Data would be available upon reasonable request from the corresponding author.

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
