# Peer review of "Oregano (Origanum vulgare) Essential Oil and Its Constituents Prevent Rat Kidney Tissue Injury and Inflammation Induced by a High Dose of L-Arginine"

_ijms, 2024, doi:10.3390/ijms25020941_

Round 1

Reviewer 1 Report (Previous Reviewer 1)

Comments and Suggestions for Authors

Ok this is fine this way.

Author Response

Thank you for the comment

Reviewer 2 Report (New Reviewer)

Comments and Suggestions for Authors

Paper is well written presenting interesting data.

Some issues that must be corrected:

1. Add film thickness of the column (I guess it was 0,25 um);

2. Is a detail composition published in Stojanović, N.M.; Mladenović, M.Z.; Maslovarić, A.; Stojiljković, N.I.; Randjelović, P.J.; Radulović, N.S. Lemon balm (Melissa 776 officinalis L.) essential oil and citronellal modulate anxiety-related symptoms - in vitro and in vivo studies. J Ethnopharmacol 2022, 777 284, 114788?

I can't find it. I advice to add the Table to this paper.

3. Sometimes commercially available EOs are adulterated with e.g. oils. They are not visible on GC-MS. Did authors performed a NMR analysis for it?

4. Re-check "L-arginine in a dose of 3.5 g/kg," L. 527;

5. Why in group OR 50 mg/kg was used, whereas carvacrol or thymol was administrated in dose 10 mg/kg? What is a real content of C and T in OR?

6. Why authors administrated OEs or pure thymol/carvacrol in a different volume of olive oil in comparision to group I (OO);

7. Increase quality of scale on Fig. 5;

8. 

Author Response

Dear Editor,

I am resubmitting the manuscript (ijms-2801607) entitled: “Oregano (Origanum vulgare) essential oil and its constituents prevent rat kidney tissue injury and inflammation induced by a high dose of L-arginine“ for consideration of publication in the International Journal of Molecular Sciences after very minor comments suggested by reviewers have been addressed.

If any further matter needs to be resolved, please, do not hesitate to contact me.

Sincerely,

Corresponding author,

Nikola M. Stojanović, MD,

Paper is well written presenting interesting data.

Answer: Thank you for your comment and kind words about our paper.

Some issues that must be corrected:

  1. Add film thickness of the column (I guess it was 0,25 um);

Answer: Indeed, the film thickness was 0.25 um which was omitted by accident in the text.

  1. Is a detail composition published in Stojanović, N.M.; Mladenović, M.Z.; Maslovarić, A.; Stojiljković, N.I.; Randjelović, P.J.; Radulović, N.S. Lemon balm (Melissa 776 officinalis L.) essential oil and citronellal modulate anxiety-related symptoms - in vitro and in vivo studies. J Ethnopharmacol 2022, 777 284, 114788?

I can't find it. I advice to add the Table to this paper.

Answer: The table with detailed composition has been previously added to the supplementary material of this work, however, during resubmission we might have forgotten to upload it again. We have corrected this now. The reference was there in order to highlight how the analysis was done.

  1. Sometimes commercially available EOs are adulterated with e.g. oils. They are not visible on GC-MS. Did authors performed a NMR analysis for it?

Answer: Yes, NMR analysis of the essential oil was performed and no peaks originating from non-volatile components were observed.

  1. Re-check "L-arginine in a dose of 3.5 g/kg," L. 527;

Answer: The dose is 350 mg/100g of bw or 3.5 g/kg, we have corrected this to state 3.5 g/kg of bw so there would be no confusion.

  1. Why in group OR 50 mg/kg was used, whereas carvacrol or thymol was administrated in dose 10 mg/kg? What is a real content of C and T in OR?

Answer: The doses of C and T were chosen based on some previous studies conducted by our research group publications and pilot experiments and based on the data from the literature. The content of C in the EO is ca 46%, while the content of T is 2.5% this can be seen from the table in supplementary material.

  1. Why authors administrated OEs or pure thymol/carvacrol in a different volume of olive oil in comparision to group I (OO);

Answer: All test substances (EO and pure compounds) were dissolved in olive oil and given to animals in the same volume as pure OO ie in a dose of 10 ml/kg. If there is some mistake in the text stating otherwise please do point us so we could correct it.

  1. Increase quality of scale on Fig. 5;

Answer: The scale given in Fig 5 is of adequate quality but due to a large number of images which are part of this figure might appear to be of poor quality. However, if you would to zoom the text you would clearly see that the scale and scale bar legend is of adequate quality and visibility. If we would to increase the scale size it would cover the structures of the tissue which is at least according to our opinion more important to be presented.

This manuscript is a resubmission of an earlier submission. The following is a list of the peer review reports and author responses from that submission.

Round 1

Reviewer 1 Report

Comments and Suggestions for Authors

Thank you for inviting me to review this original work.

In this work, the authors assess the efficacy of a treatment with Origanum vulgare essential oil or some of its compounds, in reducing the kidney damages in an acute pancreatitis rat model induced by intra-peritoneal L-Arginine injection.

The strengths of this manuscript are in the concept of the study, the introduction is well written, and the protocol is well designed.

Unfortunately, I find the presented results disappointing with out clear acute kidney injury induced. Some results are presented on inflammation markers and ROS production, but strong kidney injury markers are lacking, creatinine is not enough elevated to conclude to acute kidney injury to my point of view, and histologic results shown do not show relevant kidney lesions.

The detailed comments are as follow (please note that due to the lack of convincing results, I did not read the discussion).

Major comments:

Abstract:

-          Methods could be described shorter to let more place to the results with a bit more of details.

Material and methods:

-          The induction of the acute pancreatitis in you rat model is not proven. Did you measure pancreatic enzymes in the blood of the animals?

Results:

-          Table 1: it is difficult to understand why the rats treated with AT AC ATC and AA present decreased levels of serum creatinine compared to controls without pancreatitis. Weather this result is associated with an increase in glomerular filtration, to an increased tubular secretion of creatinine, or to a technical issue is not clear. But it does not really show that kidneys are protected from AKI related to AP. Is it possible that compounds like AT interfere with the AU860 analysis of creatinine level?

-          Renal lesion do not seem to be strong in your model, tissue KIM1 and NGAL protein level seem to be low and plasma creatinine is not so much elevated. Did you measure urinary KIM1?

-          Figure 4 is not readable, please label the picture in another way.

-          Figure 4, on the pictures shown, concerning H&E staining, G and H kidneys are injured, C and J are possibly mildly injured. The other histology samples show healthy kidneys, including F that should be your most injured sample according to your hypothesis, and what is mentioned in the text is wrong, F is healthy on the shown pictures.

-          Figure 4, there is a mistake in the scales all the magnifications are not the same.

-          Figure 5, you should score and quantify the results of the immunohistochemistry histology. This technique is way to variable to draw conclusions from so small samples of pictures.

Minor comments:

Introduction:

-          Possibly too much self-citation in the first 7 references. Please cite a review on acute pancreatitis from a high impact factor journal.

-          Line 74, the reference number 16 is wrong, it’s about a cisplatine nephrotoxic injury, not doxorubicin cardiomyopathy.

Material and methods:

-          Please provide the ages of the rats used.

Results:

-          Olive Oil should not be considered as a vehicle, but rather to a control (table 1).

-          Please notify clearly in the results that the blood samples are collected at 24h after injury. This is important because acute kidney injury usually recovers in non-lethal models.

Author Response

Dear Editor,

I am resubmitting the manuscript entitled: “Oregano (Origanum vulgare) essential oil and its constituents prevent rat kidney tissue injury and inflammation induced by a high dose of L-arginine“ for consideration of publication in the International Journal of Molecular Sciences after comments suggested by reviewers have been addressed.

The manuscript has been amended according to all of the reviewers’ comments and suggestions. All the changes made to the text are highlighted in yellow throughout the manuscript document. Point-by-point answers to specific reviewers’ comments are given below.

If any further matter needs to be resolved, please, do not hesitate to contact me.

Sincerely,

Corresponding author,

Nikola M. Stojanović, MD,

Reviewer(s)' Comments to Author:

Major comments:

Abstract:

-Methods could be described shorter to let more place to the results with a bit more of details.

Answer: We have extended the section Abstract focusing more on the obtained results.

Material and methods:

-The induction of the acute pancreatitis in you rat model is not proven. Did you measure pancreatic enzymes in the blood of the animals?

Answer: In the section Material and methods the determination of the activities of alpha-amylase, lipase and LDH enzymes in rat serum was added and in the results section these data were added.

Results:

-Table 1: it is difficult to understand why the rats treated with AT AC ATC and AA present decreased levels of serum creatinine compared to controls without pancreatitis. Weather this result is associated with an increase in glomerular filtration, to an increased tubular secretion of creatinine, or to a technical issue is not clear. But it does not really show that kidneys are protected from AKI related to AP. Is it possible that compounds like AT interfere with the AU860 analysis of creatinine level?

Answer: Indeed the results might be coming from the alterations in the kidney function, which are either through glomerular filtration or through an increased tubular secretion. We mentioned this in the discussion section now. Regarding the technical issue the method is the determination of creatinine was a routine laboratory one, and we do not know if it interferes with the method. One of the possibilities is that L-arginine could interfere with creatinine since the dose applied is quite large, however in the AOO group the creatinine values were increased compared to the treatment AT, AC, ATC and AA. The dose of the test substance is quite low and 24h is more than sufficient to be metabolized, however we could not exclude this possibility completely.

-Renal lesion do not seem to be strong in your model, tissue KIM1 and NGAL protein level seem to be low and plasma creatinine is not so much elevated. Did you measure urinary KIM1?

Answer: The results regarding KIM1 and NGAL were determined in kidney tissue, while urea and creatinine in serum. We did not analyze urine KIM1 levels. You are right, AKI associated with L-arginine and acute pancreatitis is not pronounced but rather moderate one.

-Figure 4 is not readable, please label the picture in another way.

Answer: We do understand that it is hard to read the images part of figure 4, however, we gave those images only as a conformation of the scoring system given in the Table and detailed description in the results text and bellow in the figure legend. It is quite hard to present 11 images per staining as part of a figure montage, that is why the images are small and hard to read. We still wanted every group to be present in the figure, and all staining to be shown, since it is important to present all the results that we obtained, but at the same time not to burden the text/manuscript with too many figures. If the Editor deems necessary to change this we will alter as it is suggested.

-Figure 4, on the pictures shown, concerning H&E staining, G and H kidneys are injured, C and J are possibly mildly injured. The other histology samples show healthy kidneys, including F that should be your most injured sample according to your hypothesis, and what is mentioned in the text is wrong, F is healthy on the shown pictures.

Answer: Indeed, the images C and J are mildly injured which is given in the scoring table with the results pointing to the presence of tubular casts and cloudy swelling. Also, there is a mix up with the images during the collage making, now this is corrected, image G with prominent tubular necrosis and degeneration, as well as tubular cast occurrence is image F as it should be.

-Figure 4, there is a mistake in the scales all the magnifications are not the same.

Answer: There has been a mistake in the scales, the magnifications are not the same, that is why the scales should be different, in the revised version the magnification was added for each staining and new scales introduced.

-Figure 5, you should score and quantify the results of the immunohistochemistry histology. This technique is way to variable to draw conclusions from so small samples of pictures.

Answer: Quantification results are added to the Table 3 and introduced in the results section.

Minor comments:

Introduction:

-Possibly too much self-citation in the first 7 references. Please cite a review on acute pancreatitis from a high impact factor journal.

Answer: The reference is changed and the paper showing this experimental model introduced back in early 2000s is added.

-Line 74, the reference number 16 is wrong, it’s about a cisplatine nephrotoxic injury, not doxorubicin cardiomyopathy.

Answer: Thank you for this suggestion. In the new version of Manuscript we have corrected this mistake.

Material and methods:

-Please provide the ages of the rats used.

Answer: Thank you for this suggestion. In the new version of Manuscript we have provided this information.

Results:

-Olive Oil should not be considered as a vehicle, but rather to a control (table 1).

Answer: Thank you for this constructive suggestion. In the new version of Manuscript, we have introduced this change as you suggested. 

-Please notify clearly in the results that the blood samples are collected at 24h after injury. This is important because acute kidney injury usually recovers in non-lethal models.

Answer: Thank you for this suggestion. Additionally, in the new version of manuscript we have provided this very important information.

Reviewer 2 Report

Comments and Suggestions for Authors

The manuscript is another prove of the benefits of the oregano oil and in particular thymol and carvacrol in anti-inflammatory processes.

However, some details need to add to improve the manuscript. The characteristics about the oregano oil utilized in this research are missed. Is there any reference for oregano oil in Europe? I was looking for the table S1 in the section 2.1 where is it? A complete analysis is required.

The benefits of thymol and carvacrol has been heavily published. The manuscript needs better discussion about those topics for example PMID: 37417222; 37760962. 

Other than that, the manuscript is good enough to be accepted with minor changes.

Comments on the Quality of English Language

Minor grammatical mistakes, which can be check in the next steps.

Author Response

Dear Editor,

I am resubmitting the manuscript entitled: “Oregano (Origanum vulgare) essential oil and its constituents prevent rat kidney tissue injury and inflammation induced by a high dose of L-arginine“ for consideration of publication in the International Journal of Molecular Sciences after comments suggested by reviewers have been addressed.

The manuscript has been amended according to all of the reviewers’ comments and suggestions. All the changes made to the text are highlighted in yellow throughout the manuscript document. Point-by-point answers to specific reviewers’ comments are given below.

If any further matter needs to be resolved, please, do not hesitate to contact me.

Sincerely,

Corresponding author,

Nikola M. Stojanović, MD,

Reviewers’ comments

The manuscript is another prove of the benefits of the oregano oil and in particular thymol and carvacrol in anti-inflammatory processes.

However, some details need to add to improve the manuscript. The characteristics about the oregano oil utilized in this research are missed. Is there any reference for oregano oil in Europe? I was looking for the table S1 in the section 2.1 where is it? A complete analysis is required.

Answer: Thank you for your comment. The oregano oil is available on the market and that is the way we obtained it. Unfortunately, we missed to attach the table for supplementary material with detailed oil analysis during initial submission, and we did it this time.

The benefits of thymol and carvacrol has been heavily published. The manuscript needs better discussion about those topics for example PMID: 37417222; 37760962. 

Answer: The references are added to the discussion.

Other than that, the manuscript is good enough to be accepted with minor changes

Answer: Minor grammatical and typographical errors have been mended.

Reviewer 3 Report

Comments and Suggestions for Authors

The ms is well structured and focus point are well chosen and the research approach is suitable to address the raised questions. The oregano literature is fairly large, but comparative studies in which the oregano is included are limited so the current ms is comprehensive and tries to highlight the oregano oil associated health claims in the light of major constituents. This is a novel research approach and represents a substantial added value. 

Author Response

Dear Editor,

I am resubmitting the manuscript entitled: “Oregano (Origanum vulgare) essential oil and its constituents prevent rat kidney tissue injury and inflammation induced by a high dose of L-arginine“ for consideration of publication in the International Journal of Molecular Sciences after comments suggested by reviewers have been addressed.

The manuscript has been amended according to all of the reviewers’ comments and suggestions. All the changes made to the text are highlighted in yellow throughout the manuscript document. Point-by-point answers to specific reviewers’ comments are given below.

If any further matter needs to be resolved, please, do not hesitate to contact me.

Sincerely,

Corresponding author,

Nikola M. Stojanović, MD,

Reviewers’ comments

The ms is well structured and focus point are well chosen and the research approach is suitable to address the raised questions. The oregano literature is fairly large, but comparative studies in which the oregano is included are limited so the current ms is comprehensive and tries to highlight the oregano oil associated health claims in the light of major constituents. This is a novel research approach and represents a substantial added value. 

Answer: Dear Reviewer thank you for your comments.

Round 2

Reviewer 1 Report

Comments and Suggestions for Authors

What I understand from these update version is that control groupe with olive oil, present probably acute pancreatitis, as shown by the elevated lipase level.

Whether L-arginine alone induces acute pancreatitis is not clear. 

Furthermore, the studied treatment does probably not aleviate kidney lesions specifically, but blunts pancreatitis severity.

The all article should be written in this way but the scientific soundness is very attenuated regarding this point.

Finaly, the scales of the pictures have been changed, to me the pictures should have been changed to be homogeneous.

Author Response

Dear Editor,

I am resubmitting the manuscript entitled: “Oregano (Origanum vulgare) essential oil and its constituents prevent rat kidney tissue injury and inflammation induced by a high dose of L-arginine“ for consideration of publication in the International Journal of Molecular Sciences after comments suggested by reviewers have been addressed.

If any further matter needs to be resolved, please, do not hesitate to contact me.

Sincerely,

Corresponding author,

Nikola M. Stojanović, MD,

What I understand from these update version is that control groupe with olive oil, present probably acute pancreatitis, as shown by the elevated lipase level.

Answer: Yes, as in all previous experiments conducted by our research group and published papers with the model involving acute pancreatitis, the control group is the one receiving olive oil and L-arginine. The induction of acute pancreatitis is not only seen through lipase activity, but through a significant increase in amylase and LDH activities as well.

Whether L-arginine alone induces acute pancreatitis is not clear.

Answer: The authors understand what the reviewer wants to say, but as in previous publications from our research group and in the ones cited, injection of a high dose of L-arginine causes massive pancreatic damage. This is always visible through the enzymes in serum and pancreatic and extra-pancreatic tissue changes.

Furthermore, the studied treatment does probably not aleviate kidney lesions specifically, but blunts pancreatitis severity.

Answer: This kind of doubt and uncertainty is the activity arriving from preventing pancreatic damage or from preventing direct damage of L-arginine to kidneys is discussed through the manuscript. The most proper answer would be that the activity arises from both actions, since it is undoubtedly to high dose of L-arginine to be handled safely by kidneys, or any tissue for that matter.

The all article should be written in this way but the scientific soundness is very attenuated regarding this point.

Answer: The manuscript clearly stated both facts, acute pancreatic damage and kidney damage arising from L-arginine application. Both introduction and discussion are written in that manner. If the reviewer sees the place for improvement please do clearly suggest what part should be rewritten to make it more to the point. Our conclusions clearly state that the observed activity comes from ‘‘…prevent L-arginine-induced tissue damage and potentially improve the prognosis of pancreatitis by reducing damage to distant organs.’’

Finaly, the scales of the pictures have been changed, to me the pictures should have been changed to be homogeneous.

Answer: Thank you for your comment, we patriciate it.

Round 3

Reviewer 1 Report

Comments and Suggestions for Authors

Furthermore, the studied treatment does probably not aleviate kidney lesions specifically, but blunts pancreatitis severity.

Answer: This kind of doubt and uncertainty is the activity arriving from preventing pancreatic damage or from preventing direct damage of L-arginine to kidneys is discussed through the manuscript. The most proper answer would be that the activity arises from both actions, since it is undoubtedly to high dose of L-arginine to be handled safely by kidneys, or any tissue for that matter.

=> To my point of view, it should be clearly mentionned that it is really more likely that the kidney protective effect is due to the diminished intensity of the pancreatitis and not to a specific kidney effect.